# Association Between Chewing Status and Steatotic Liver Disease in Japanese People Aged ≥50 Years: A Cohort Study

**DOI:** 10.3390/healthcare13121399

**Published:** 2025-06-11

**Authors:** Komei Iwai, Daisuke Ekuni, Tetsuji Azuma, Takatoshi Yonenaga, Koichiro Tabata, Naoki Toyama, Kota Kataoka, Takayuki Maruyama, Takaaki Tomofuji

**Affiliations:** 1Department of Community Oral Health, School of Dentistry, Asahi University, 1-1851 Hozumi, Mizuho 501-0296, Japan; ko-mei@dent.asahi-u.ac.jp (K.I.); tetsuji@dent.asahi-u.ac.jp (T.A.); yone0730@dent.asahi-u.ac.jp (T.Y.); tabata-k@dent.asahi-u.ac.jp (K.T.); 2Department of Preventive Dentistry, Okayama University Graduate School of Medicine, Dentistry and Pharmaceutical Sciences, 2-5-1 Shikata-cho, Kita-ku, Okayama 700-8558, Japan; dekuni7@md.okayama-u.ac.jp (D.E.); pu171qxi@s.okayama-u.ac.jp (N.T.); de18017@s.okayama-u.ac.jp (K.K.); t-maru@md.okayama-u.ac.jp (T.M.)

**Keywords:** oral health, liver diseases, longitudinal studies, mastication, physical examination, surveys and questionnaires

## Abstract

**Background/Objectives:** In this longitudinal study, the relationship between chewing status and steatotic liver disease (SLD) was examined in 3775 people aged ≥50 years who underwent medical checkups at Junpukai Health Maintenance Center in Okayama, Japan. **Methods:** Participants without SLD at the time of a baseline survey in 2018 were followed until 2022. Chewing status was assessed by a self-administered questionnaire. The presence or absence of SLD was ascertained from the medical records of Junpukai Health Maintenance Center. **Results:** A total of 541 participants (14%) were diagnosed as having a poor chewing status at baseline. Furthermore, 318 (8%) participants were newly diagnosed with SLD at follow-up. In multivariate logistic regression analyses, the presence or absence of SLD was found to be associated with the following characteristics at baseline: sex (male: odds ratio [ORs] = 1.806; 95% confidence interval [CIs]: 1.399–2.351), age (ORs = 0.969; 95% CIs: 0.948–0.991), body mass index (≥25.0 kg/m^2^; ORs = 1.934; 95% CIs: 1.467–2.549), diastolic blood pressure (ORs = 1.017; 95% CIs: 1.002–1.032), and chewing status (poor: ORs = 1.472; 95% CIs: 1.087–1.994). **Conclusions:** The results indicate that a poor chewing status was associated with SLD development after 4 years. Aggressively recommending dental visits to participants with poor chewing status may not only improve their ability to chew well but may also reduce the incidence of SLD.

## 1. Introduction

Steatotic liver disease (SLD) is the overarching term for conditions in which excess fat accumulates in the liver and lipolysis is impaired, resulting in abnormal lipid metabolism and an unhealthy condition in the liver [1,2]. In recent years, the proportion of Japanese patients with SLD has been increasing, and it is reported that about 10–30% of Japanese people have SLD [3,4,5,6]. Patients with SLD often show few subjective symptoms when the disease is mild [7]. However, as symptoms progress, liver failure, kidney failure, and liver cancer may occur [8,9,10]. Therefore, identifying and controlling factors associated with new-onset SLD will contribute to preventing the development of liver failure, kidney failure, and liver cancer.

Chewing status is one of the health conditions related to eating behavior. In a clinical study, it was shown that chewing probiotics-based gum could improve periodontal health [11]. Another clinical study reported that poor chewing status is a risk factor for dental caries and molar malocclusion [12]. In this way, it is well known that chewing is associated with oral health.

It has also been shown that chewing is implicated in fat accumulation in the body. For example, past epidemiological studies showed that chewing status is associated with abdominal obesity [13,14]. Furthermore, another epidemiological study also reported the association between poor chewing status and SLD [15]. However, these studies showed the cross-sectional association between chewing status and fat accumulation status, and it is not still clear whether poor chewing status could lead to SLD in the future.

In Japan, the Ministry of Health, Labour and Welfare mandates the provision of specific health checkups focusing on lifestyle [16,17]. The questionnaire for these specific health checkups includes items such as eating behavior, including chewing status [18]. Furthermore, in Junpukai Health Maintenance Center in Okayama, Japan, participants who undergo these health checkups can receive medical checkups for SLD if they so desire. Therefore, by combining these data, the association between chewing status and SLD can be investigated.

In addition, it has been reported that chewing status begins to decline around middle age [19], and in Japan, an increase in the “proportion of Japanese people aged ≥50 years with good chewing status” has been set as a health principle in the national health promotion movement “Healthy Japan 21” to promote the health of Japanese people in the 21st century. In other words, there is a high possibility of finding a longitudinal association between masticatory status and SLD in a population aged ≥50 years, in which reduced mastication is more likely to be observed. Given this background, the aim of the present longitudinal study was to clarify the longitudinal relationship between chewing status and SLD in Japanese people aged ≥50 years who had undergone medical checkups at Junpukai Health Maintenance Center over a 4-year period.

## 2. Materials and Methods

### 2.1. Study Design

This was a prospective cohort study with a follow-up period of 4 years. The disease of interest was SLD, and the factor considered was good or poor chewing status.

### 2.2. Participants

Our study comprises an all-participant survey of community residents who underwent medical checkups at Junpukai Health Maintenance Center in Okayama, Japan, and who were not diagnosed with SLD in the 2018 baseline survey. Informed consent was not obtained because this study used anonymized claims data. In total, 10,411 Japanese people aged ≥50 years participated in the baseline survey, which was conducted between April 2018 and March 2019. At baseline, 2458 individuals with SLD and 170 with missing blood test data were excluded from the analysis. Of the remaining 7783 participants, 3775 were followed from April 2022 to March 2023 (follow-up rate: 49%). Therefore, in the present study, data from 3775 community-dwelling residents (1620 male, 2155 female; mean age, 57.6 years) were analyzed. A total of 318 (8%) participants were newly diagnosed with SLD at follow-up (Figure 1).

### 2.3. Assessment of SLD

The presence or absence of SLD was diagnosed by abdominal ultrasonography at Junpukai Health Maintenance Center. A skilled technician performed real-time ultrasound examinations to detect blurred vessels, deep attenuation, and increased echotexture of the liver compared with the kidneys to assess the degree of fatty infiltration [20]. An expert physician then confirmed the validity of the findings.

### 2.4. Assessment of Body Composition

Nurses measured the height and weight of participants. The body mass index (BMI) was calculated as the weight divided by the square of the height (kg/m^2^) [15]. In general, BMI ≥ 25.0 kg/m^2^ indicates obesity in Japan [21]. Therefore, the presence or absence of BMI ≥ 25.0 kg/m^2^ was analyzed as a factor in this study.

### 2.5. Assessment of the Serum Hemoglobin A1c (HbA1c) Level, Triglyceride Level, and High-Density Lipoprotein (HDL) Cholesterol Level

High-performance liquid chromatography was used to measure serum HbA1c levels in venous blood samples collected after overnight fasting [22]. Serum HbA1c levels ≥ 6.5% are generally considered to indicate poor glycemic control [23]. Therefore, the presence or absence of a serum HbA1c level ≥ 6.5% was analyzed as a factor in this study. Therefore, triglyceride level ≥ 150 mg/dL or HDL cholesterol level ≤ 40 mg/dL are generally considered two of the diagnostic indices for dyslipidemia [24]. Therefore, the presence or absence of triglyceride level ≥ 150 mg/dL or HDL cholesterol level ≤ 40 mg/dL was analyzed as a factor in this study.

### 2.6. Assessment of Blood Pressure Levels

Nurses measured the systolic and diastolic blood pressures of the participants. Blood pressure levels were measured twice for each participant, and the mean value was calculated [25].

### 2.7. Assessment of Chewing Status and Other Items by a Self-Administered Questionnaire

The self-administered questionnaire was the same one used in specific health checkups in Japan [26]. Regarding current chewing status, participants selected from: “I can eat anything.”; “I sometimes have difficulty chewing due to dental problems such as dental caries or periodontal disease.”; and “I can hardly chew.” Participants who answered “I can eat anything.” were defined as having a good chewing status, whereas those who answered “I sometimes have difficulty chewing due to dental problems such as dental caries or periodontal disease.” and “I can hardly chew.” were defined as having poor chewing status [13]. The questionnaire also included items on the following variables: sex, age, smoking status (the presence or absence of smoking at least one cigarette per day), drinking status (the presence or absence of drinking alcohol at least once per day), exercise habits (the presence or absence of engaging in light exercise for at least 30 min more than twice per week for at least 1 year), physical activity (the presence or absence of going for a walk or the equivalent for at least 1 h per day), sleep status (good or poor), eating speed (slow, medium, or quick), snacking habits (none, sometimes, or daily), skipping breakfast (<3 or ≥3 times/week), and eating dinner within 2 h before bedtime (<3 or ≥3 times/week) [16,26,27,28].

### 2.8. Statistical Analysis

The normality of the data was checked with the Lilliefors tests. Because the results indicated that the continuous variables were not all normally distributed, the data were expressed as median (first- and third-quartile) values. Fisher’s exact test and the Mann–Whitney *U* test were used to assess significant differences in the characteristics of each factor by chewing status (good or poor). In addition, univariate and multivariate logistic regression analyses were performed using the presence of SLD as the dependent variable. In the multivariate stepwise logistic regression analysis, factors that were significantly different in the univariate logistic regression analysis in addition to sex and age were selected for the third category. The Hosmer–Lemeshow fit test was performed to confirm the suitability of the model using multivariate stepwise logistic regression analysis. All data were analyzed using SPSS (version 27; IBM Japan, Tokyo, Japan), with a *p*-value < 0.05 considered to indicate statistical significance.

### 2.9. Research Ethics

The Ethics Committee of Asahi University approved the present study (approval Nos. 27010 and 30018), which was performed in accordance with the Declaration of Helsinki (as revised in Brazil in 2013).

## 3. Results

Table 1 shows the characteristics of the participants by chewing status at baseline. A total of 541 participants (14%) were diagnosed as having a poor chewing status at baseline. Participants with a poor chewing status were characterized by a significantly higher proportion of males (*p* < 0.001), smoking (*p* < 0.001), drinking (*p* < 0.001), poor sleep status (*p* < 0.001), skipping breakfast ≥ 3 times/week (*p* < 0.001), and dinner within 2 h before bedtime ≥ 3 times/week (*p* < 0.001). Furthermore, participants with a poor chewing status were characterized by a significantly lower age (*p* = 0.024), BMI (*p* < 0.001), triglyceride level (*p* < 0.001), systolic blood pressure (*p* < 0.001), and diastolic blood pressure (*p* < 0.001).

Table 2 shows the relationship between chewing status at baseline and SLD at follow-up. Participants with a poor chewing status at baseline were characterized by a significantly higher proportion with SLD at follow-up (*p* = 0.002).

The crude odds ratios (ORs) and 95% confidence intervals (CIs) for SLD at follow-up are shown in Table 3. As shown in the table, the risk of SLD after 4 years was significantly correlated with the following variables at baseline: sex (male: ORs = 2.165; 95% CIs: 1.712–2.738), age (ORs = 0.968; 95% CIs: 0.948–0.988), BMI ≥ 25.0 kg/m^2^ (ORs = 2.281; 95% CIs: 1.764–2.950), triglyceride level ≥ 150 mg/dL (ORs = 2.499; 95% CIs: 1.118–5.583), HDL cholesterol level ≤ 40 mg/dL (ORs = 2.131; 95% CIs: 1.523–2.983), smoking status (ORs = 1.496; 95% CIs: 1.123–1.993), systolic blood pressure (ORs = 1.007; 95% CIs: 1.001–1.013), diastolic blood pressure (ORs = 1.019; 95% CIs: 1.010–1.029), sleep status (poor: ORs = 1.278; 95% CIs: 1.013–1.613), poor chewing status (ORs = 1.574; 95% CIs: 1.177–2.105), skipping breakfast ≥ 3 times/week (ORs = 1.594; 95% CIs: 1.140–2.229), and having dinner within 2 h before bedtime ≥ 3 times/week (ORs = 1.671; 95% CIs: 1.307–2.136).

The adjusted ORs and 95% CIs for SLD at follow-up are shown in Table 4. After adjusting for sex, age, BMI, triglyceride level, HDL cholesterol level, smoking status, systolic and diastolic blood pressure, sleep status, chewing status, skipping breakfast, and having dinner within 2 h before bedtime, the risk of SLD at 4 years was significantly correlated with the following variables at baseline: sex (male: ORs = 1.806; 95% CIs: 1.399–2.351), age (ORs = 0.969; 95% CIs: 0.948–0.991), BMI ≥ 25.0 kg/m^2^ (ORs = 1.934; 95% CIs: 1.467–2.549), diastolic blood pressure (ORs = 1.017; 95% CIs: 1.002–1.032), and poor chewing status (ORs = 1.472; 95% CIs: 1.087–1.994).

## 4. Discussion

To the best of our knowledge, this is the first longitudinal study to examine the associations between chewing status and SLD in Japanese people aged ≥50 years using data from medical checkups. The results showed that a higher proportion of participants with poor chewing status at baseline had SLD within 4 years than those with good chewing status. The results of the logistic regression analysis showed that after adjusting for sex, age, BMI, triglyceride level, HDL cholesterol level, smoking, systolic blood pressure, diastolic blood pressure, sleep status, skipping breakfast, and dinner within 2 h before bedtime, the presence or absence of SLD after 4 years was associated with poor chewing status at baseline. These findings indicated that a decrease in chewing status was associated with a higher risk of developing SLD in the future.

There are several possible mechanisms behind the association between chewing status and SLD. First, a lack of systemic caloric expenditure may be involved. It has been noted that poor chewing status causes a decrease in diet-induced heat production and the induction of neurohistamine inactivation [29,30,31]. The lack of calories consumed due to these factors may affect fat accumulation in the liver. Second, eating habits may also be involved. It has been reported that people with poor chewing status tend to consume less vegetables and fruits and more soft high-energy foods than those with good chewing status [32,33,34,35]. Furthermore, it has also been also noted that an increase in the number of chews increases subjective satiety after a meal, which is involved in the prevention of overeating [36]. It has been reported that overeating affects fat accumulation in the liver [37]. These factors contribute to obesity and may affect fat accumulation in the liver. Furthermore, since estrogen has been reported to inhibit fat synthesis in the liver and lower blood fat [38], decreased production of estrogen may also be related to chewing status and SLD. The oral microflora is also known to correlate with the estrobolome of a subset of bacterial species with genes encoding β-glucuronidase and β-galactosidase [39]. Therefore, poor oral hygiene associated with poor mastication status may reduce estrogen production and be associated with the development of SLD. However, the mechanism by which poor chewing status is associated with SLD needs to be clarified in future research.

The relationship between chewing status and fat accumulation has been reported in previous studies. For example, chewing difficulty has been found to affect abdominal obesity [40]. It has also been shown that unbalanced chewing habits due to poor chewing status are associated with overweight and obesity [41]. Previous studies and the present study support the concept that poor chewing status may be involved in unhealthy fat accumulation in the body.

The results of our study suggest that an improvement in chewing status may prevent the future development of SLD and improve fat accumulation in the liver. Actively recommending dental visits to examinees with poor chewing status, treatment by orthodontists [42], and dental health guidance by dental hygienists [43] may not only improve chewing status but also prevent the onset of SLD.

In this study, a self-administered questionnaire was used to assess the participants’ chewing status. Therefore, participants’ subjective symptoms and actual chewing ability may differ. However, the self-reported chewing status is considered valid because it has been reported that it is not only related to the number of existing teeth and molars but also useful as a screening method regarding actual chewing ability using the same questionnaire [44].

In the present study, male sex, younger age (among aged ≥50 years), BMI, and diastolic blood pressure were associated with the presence of SLD in participants aged ≥50 years. These results are consistent with previous studies reporting that SLD is a disease that is more likely to affect men [6,45] and is associated with higher BMI [46,47]. In addition, a previous study reported that the most common age of people with SLD in Japan is about 52 years [48]. Therefore, it is possible that SLD tended to develop at a younger age in the present study, which was limited to participants aged ≥50 years. Furthermore, an elevated diastolic blood pressure is not only a hypertensive state but also suggests the progression of arterial stiffness [49]. SLD is associated with atherosclerosis, as well as hypertension [50,51]. Therefore, though both systolic and diastolic blood pressures were associated with SLD in the univariate analysis, diastolic blood pressure may have been associated with SLD as a stronger factor in the multivariate analysis, in which both factors were analyzed simultaneously.

The proportion of participants with SLD was 8% and that with poor chewing status was 14%. In Japan, it has been reported that the proportion of people with SLD ranges from 10% to 30% [3,4,5,6]. Furthermore, according to data from Health Japan 21 (the third term), the proportion of people aged ≥ 50 years with poor chewing status in 2019 was 29% [52]. This indicates that the proportion of participants with SLD and poor chewing status in this study was lower than that of the average Japanese population. The reason for this is that the participants were people who underwent medical checkups, suggesting that they may have been more health-conscious on a daily basis. Therefore, the results may differ if a different health population were to be targeted.

The multivariate logistic regression analysis model used in the present study utilized the Hosmer–Lemeshow fit test, which is capable of examining the fit of multivariate logistic regression analysis models and testing the compatibility of the event rate observed in a subgroup model and the expected event rate. In the Hosmer–Lemeshow fit test, a *p*-value > 0.05 is considered to indicate a good fit [53]. In the present study, the *p*-value was 0.704, thereby indicating a good fit for the multivariate logistic regression analysis model.

However, there are several limitations in the present study. First, the participants included only those who underwent medical checkups at Junpukai Health Maintenance Center. Therefore, external validity should be considered. Second, we did not investigate the amount of alcohol consumption. Since the definition of alcohol intake is important when considering liver disease, we would like to consider future investigations regarding these issues. Third, the dietary intake of the participants was not investigated. A past study reported that SLD is associated with fructose intake and unhealthy eating habits [54]. It is possible that usual eating habits may affect fat accumulation, and we would like to examine this in the future. However, a major strength of the present study is the sample size of over 3700 Japanese people aged ≥50 years. This sample size is sufficient to demonstrate a longitudinal relationship between chewing status and future SLD development, and it may help to find factors associated with unhealthy fat accumulation in the liver in the Japanese population.

## 5. Conclusions

The results of the present study indicate that Japanese people aged ≥50 years with poor chewing status are at an increased risk of developing SLD after 4 years. Therefore, to prevent unhealthy fat accumulation in the liver, it is important to maintain good chewing status. Aggressively recommending dental visits to participants with poor chewing status may not only improve their ability to chew well but may also reduce the incidence of SLD.

## Figures and Tables

**Figure 1 healthcare-13-01399-f001:**
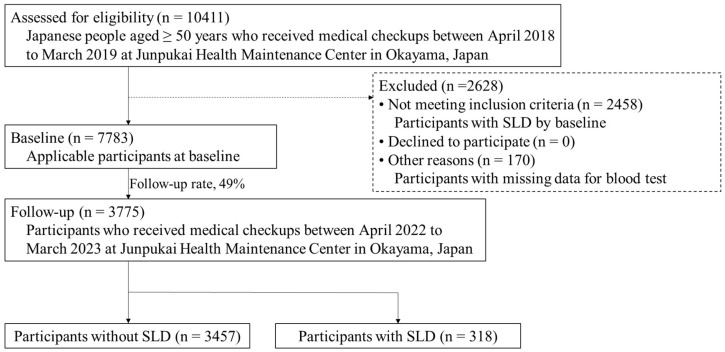
Flowchart of data selection criteria. Abbreviations: SLD, steatotic liver disease.

**Table 1 healthcare-13-01399-t001:** Differences in factors at baseline between good and poor chewing status.

Factor	Chewing Status		*p*-Value *
	Good (n = 3234)	Poor (n = 541)	
Sex ^a^	1345 (42%)	275 (50%)	<0.001 *
Age (years)	57 (53, 62)	56 (52, 62)	0.024 *
BMI (kg/m^2^)	22.0 (20.1, 24.1)	19.7 (21.6, 23.7)	<0.001 *
HbA1c level (%)	5.6 (5.5, 5.9)	5.7 (5.5, 5.9)	0.879
Triglyceride level (mg/dL)	81.0 (60.0, 114.0)	86.0 (64.8, 133.3)	<0.001 *
HDL cholesterol level (mg/dL)	72.0 (60.0, 86.0)	69.0 (57.8, 83.0)	0.655
Smoking habit ^b^	443 (14%)	138 (25%)	<0.001 *
Drinking habit ^b^	906 (28%)	198 (37%)	<0.001 *
Exercise habit ^b^	1069 (33%)	171 (32%)	0.507
Physical activity ^b^	823 (25%)	124 (23%)	0.209
Systolic blood pressure level (mmHg)	120 (107, 133)	114 (102, 128)	<0.001 *
Diastolic blood pressure level (mmHg)	75 (67, 84)	72 (65, 80)	<0.001 *
Sleep status			
Well	2056 (64%)	285 (53%)	<0.001 *
Poor	1178 (36%)	256 (47%)	
Eating speed			
Slowly	275 (8%)	61 (11%)	0.098
Medium	1925 (60%)	318 (59%)	
Quickly	1034 (32%)	162 (30%)	
Snacking habit			
None	591 (18%)	93 (17%)	0.686
Sometimes	1657 (51%)	274 (51%)	
Daily	986 (31%)	174 (32%)	
Skipping breakfast habit			
<3 times/week	2939 (91%)	467 (86%)	<0.001 *
≥3 times/week	295 (9%)	74 (14%)	
Dinner within 2 h before bedtime habit			
<3 times/week	2496 (77%)	367 (68%)	<0.001 *
≥3 times/week	738 (23%)	174 (32%)	

Abbreviations: BMI, body mass index; HbA1c, hemoglobin A1c; HDL, high-density lipoprotein. * *p*  <  0.05, using Fisher’s exact test or Mann–Whitney U test. ^a^ Male (proportion of males); ^b^ Presence (proportion of presence).

**Table 2 healthcare-13-01399-t002:** Relationship between chewing status at baseline and SLD at follow-up.

		Chewing Status at Baseline		*p*-Value *
		Good (n = 3234)	Poor (n = 541)	
**SLD at follow-up**	**Absence**	2980 (92%)	477 (88%)	0.002 *
	**Presence**	254 (8%)	64 (12%)	

Abbreviations: SLD, steatotic liver disease. * *p*  <  0.05, using Fishers exact test.

**Table 3 healthcare-13-01399-t003:** Crude ORs and 95% CIs for SLD at follow-up.

Factor		ORs	95% CIs	*p*-Value
Sex	Female	1	(reference)	<0.001
	Male	2.165	1.712–2.738	
Age (years)		0.968	0.948–0.988	0.002
BMI (kg/m^2^)	<25.0	1	(reference)	<0.001
	≥25.0	2.281	1.764–2.950	
HbA1c level (%)	<6.5	1	(reference)	0.550
	≥6.5	0.844	0.484–1.472	
Triglyceride level (mg/dL)	<150	1	(reference)	0.026
	≥150	2.499	1.118–5.583	
HDL cholesterol level (mg/dL)	>40	1	(reference)	<0.001
	≤40	2.131	1.523–2.983	
Smoking habits	Absence	1	(reference)	0.006
	Presence	1.496	1.123–1.993	
Drinking habit	Absence	1	(reference)	0.072
	Presence	1.251	0.980–1.597	
Exercise habit	Presence	1	(reference)	0.054
	Absence	1.284	0.995–1.657	
Physical activity	Presence	1	(reference)	0.360
	Absence	1.136	0.865–1.481	
Systolic blood pressure level (mmHg)		1.007	1.001–1.013	0.020
Diastolic blood pressure level (mmHg)		1.019	1.010–1.029	<0.001
Sleep status	Well	1	(reference)	0.038
	Poor	1.278	1.013–1.613	
Chewing status	Good	1	(reference)	0.002
	Poor	1.574	1.177–2.105	
Eating speed	Not quickly	1	(reference)	0.508
	Quickly	1.086	0.851–1.386	
Snacking habit	Not daily	1	(reference)	0.327
	Daily	0.881	0.683–1.136	
Skipping breakfast habit	<3 times/week	1	(reference)	0.006
	≥3 times/week	1.594	1.140–2.229	
Dinner within 2 h before bedtime habit	<3 times/week	1	(reference)	<0.001
	≥3 times/week	1.671	1.307–2.136	

Abbreviations: SLD, steatotic liver disease; ORs, odds ratios; CIs, confidence intervals; BMI, body mass index; HbA1c, hemoglobin A1c; HDL, high-density lipoprotein.

**Table 4 healthcare-13-01399-t004:** Adjusted ORs and 95% CIs for SLD at follow-up.

Factor		ORs	95% CIs	*p*-Value
Sex	Female	1	(reference)	<0.001
	Male	1.806	1.399–2.351	
Age (years)		0.969	0.948–0.991	0.005
BMI (kg/m^2^)	<25.0	1	(reference)	<0.001
	≥25.0	1.934	1.467–2.549	
Diastolic blood pressure level (mmHg)		1.017	1.002–1.032	0.029
Chewing status	Good	1	(reference)	0.012
	Poor	1.472	1.087–1.994	

Abbreviations: SLD, steatotic liver disease; ORs, odds ratios; CIs, confidence intervals; BMI, body mass index; HDL, high-density lipoprotein. Adjustment for sex, age, BMI, triglyceride level, HDL cholesterol level, smoking habits, systolic blood pressure level, diastolic blood pressure level, sleep status, chewing status, skipping breakfast habit, and dinner within 2 h before bedtime habit. Hosmer–Lemeshow fit test; *p* = 0.704.

## Data Availability

The data presented in this study are available on request from the corresponding author. The data are not publicly available due to ethical restrictions.

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
