# Peer review of "Association Between Chewing Status and Steatotic Liver Disease in Japanese People Aged ≥50 Years: A Cohort Study"

_healthcare, 2025, doi:10.3390/healthcare13121399_

Round 1
Reviewer 1 Report
Comments and Suggestions for Authors
The authors described that a poor chewing state was associated with fatty liver development in Japanese people aged ≥ 50 years who underwent medical heath checkup. This manuscript is interesting; however, a crucial point requires revision.
- In Introduction, the authors seemed to mention about the epidemiology of non-alcoholic fatty liver disease (NALFD); however, it is unclear whether “fatty liver” in this manuscript refers to NAFLD. This point is crucial and should be clear. In addition, recently, NAFLD has been redefined as metabolic dysfunction-associated steatotic liver disease (MASLD) (Rinella ME, et al. J Hepatol 2023), so the authors evaluate fatty liver based on the criteria of MASLD.
- The authors described drinking status; however, it is difficult to distinguish between MASLD or non-MASLD from the criteria of drinking status (present or absent of drinking alcohol at least once per day). The amount of alcohol consumption should be clearly evaluated and described.
- If this manuscript will refer to MASLD, presence or absence of dyslipidemia should be evaluated, because a recent study demonstrated that presence of dyslipidemia is associated with MASLD onset in individuals without steatosis (Tsutsumi T, et al. J Gastroenterol. 2025).
- Can improvement of chewing state contribute to the amelioration of MASLD? Please describe this point in Discussion.
Minor
- More recently, Fujii H, et al. have reported the epidemiology of NAFLD in the general population from 2014 to 2018 in Japan (Fujii H. et al. Hepatol Res. 2023).
Author Response
In Introduction, the authors seemed to mention about the epidemiology of non-alcoholic fatty liver disease (NALFD); however, it is unclear whether “fatty liver” in this manuscript refers to NAFLD. This point is crucial and should be clear. In addition, recently, NAFLD has been redefined as metabolic dysfunction-associated steatotic liver disease (MASLD) (Rinella ME, et al. J Hepatol 2023), so the authors evaluate fatty liver based on the criteria of MASLD.
Response: We thank the reviewer for your valuable advice. We classify diseases only by presence or absence of participants with fatty deposits of ≥ 5% in liver using abdominal ultrasonography. In addition, we did not examine blood glucose levels within 2 hours after a meal in participants. Therefore, unfortunately, we can determine up to steatotic liver disease (SLD), but it is difficult to determine fatty liver disease associated with metabolic dysfunction (MASLD). Based on your suggestion, we have changed the sentence from fatty liver to SLD, as this may confuse the readers (lines 2, 16, 18, 19, 22, 23, 27, 30, 35, 38, 39, 42, 48, 50, 55, 56, 63, 66, 72, 77, 80, 85, 89, 90, 142, 165, 167, 170, 171, 183, 186, 195, 197, 201, 203, 205, 216, 220, 221, 229, 232, 240, 241, 243, 244, 247, 249, 252, 254, 256, 273, 277, 282, 286, Reference 1).
The authors described drinking status; however, it is difficult to distinguish between MASLD or non-MASLD from the criteria of drinking status (present or absent of drinking alcohol at least once per day). The amount of alcohol consumption should be clearly evaluated and described.
Response: We thank the reviewer for your valuable advice. In our study, no classification by alcohol was made with regard to SLD. Therefore, we did not investigate the amount of alcohol consumption. However, since the definition of alcohol intake is important when considering liver disease, we would like to consider future investigations regarding these issues. We have added the text to “limitations” (lines 269-272).
If this manuscript will refer to MASLD, presence or absence of dyslipidemia should be evaluated, because a recent study demonstrated that presence of dyslipidemia is associated with MASLD onset in individuals without steatosis (Tsutsumi T, et al. J Gastroenterol. 2025).
Response: We thank the reviewer for your valuable advice. We do not have information on the presence or absence of participants with dyslipidemia. However, we have obtained information on high density lipoprotein (HDL) cholesterol level and triglyceride level, which are the diagnostic indices for dyslipidemia. Therefore, based on your suggestion, we have added these factors and reanalyzed the data (lines 24-27, 107-111, 161-162, 174-175, 184, 187-190, 199, Table 1, 3, 4, Reference 22).
Can improvement of chewing state contribute to the amelioration of MASLD? Please describe this point in Discussion.
Response: We thank the reviewer for your valuable advice. As you indicated, the results of our study suggest that improvement of chewing status may prevent the future development of SLD and improve fat accumulation in the liver. We have added the point and references in the Discussion section according to your suggestion (lines 228-232, Reference 40-41).
More recently, Fujii H, et al. have reported the epidemiology of NAFLD in the general population from 2014 to 2018 in Japan (Fujii H. et al. Hepatol Res. 2023).
Response: We thank the reviewer for your valuable advice. We have added “References” according to your suggestion (lines 39, 242, 254, Reference 6).
Reviewer 2 Report
Comments and Suggestions for Authors
This is a well-structured and clearly written manuscript. The aim of the present longitudinal study is to clarify the longitudinal relationship between chewing status and fatty liver in Japanese people aged ≥ 50 years who had undergone medical checkups at Junpukai Health Maintenance Center over a 4-year period. The study is relevant to both clinical and public health fields, particularly in light of the growing recognition of oral-systemic health interactions.
The introduction provides adequate background, and the rationale for the study is well-articulated.
The methodology is very good and described in sufficient detail to allow replication. The statistical analyses are appropriate for the data and the hypothesis.
The manuscript includes a clear explanation of the study population, inclusion/exclusion criteria, assessment methods, and statistical procedures, which supports reproducibility. The authors' use of established clinical criteria for fatty liver diagnosis and adjustment for relevant confounders strengthens the credibility of the findings.
Tables and figures are relevant, well-organized, and easy to interpret.
The conclusions are well-supported by the results and do not overreach. Also, the authors appropriately discuss the clinical implications of their findings and the study’s limitations.
The authors states in the manuscript that chewing status was assessed through a self-reported questionnaire. Since this is a subjective measure and not a clinical examination, should be added a brief note or more bibliographic references to indicate that the tool used (the questionnaire) has been employed in previous studies, or It has been validated, meaning that it has been scientifically shown to correlate well with objective methods of assessing masticatory function. If this is not available, then they could add that it represents a limitation of the study.
Author Response
The introduction provides adequate background, and the rationale for the study is well-articulated. The methodology is very good and described in sufficient detail to allow replication. The statistical analyses are appropriate for the data and the hypothesis. The manuscript includes a clear explanation of the study population, inclusion/exclusion criteria, assessment methods, and statistical procedures, which supports reproducibility. The authors' use of established clinical criteria for fatty liver diagnosis and adjustment for relevant confounders strengthens the credibility of the findings. Tables and figures are relevant, well-organized, and easy to interpret. The conclusions are well-supported by the results and do not overreach. Also, the authors appropriately discuss the clinical implications of their findings and the study’s limitations. The authors states in the manuscript that chewing status was assessed through a self-reported questionnaire. Since this is a subjective measure and not a clinical examination, should be added a brief note or more bibliographic references to indicate that the tool used (the questionnaire) has been employed in previous studies, or It has been validated, meaning that it has been scientifically shown to correlate well with objective methods of assessing masticatory function. If this is not available, then they could add that it represents a limitation of the study.
Response: Thank you for your detailed guidance from "Introduction" to "Conclusions", and the structure to “Figure” and “Tables”. In addition, as you indicated, we have used a self-administered questionnaire to ascertain the chewing status of our participants. Therefore, it is possible that the participants’ subjective symptoms and their actual chewing ability may differ. On the other hands, it has been reported that self-reported chewing status is not only related to the number of present teeth and molars, but also useful as a screening method in regard to actual chewing ability. We have added the text to “Discussion” and added “References” according to your suggestion (lines 233-238, Reference 42).
Reviewer 3 Report
Comments and Suggestions for Authors
Manuscript of considerable interest for the dental sector, which highlights the importance of chewing and intestinal and hepatic health
I think the importance of the results obtained should be emphasized in the abstract because the conclusions are a bit weak.
The keywords are few and not registered on MeSh, please update them.
In the introduction I would do an overview of all the natural substances that can regulate the estrobolome even in the form of chewing gum (add reference of the estrobolome (Orrù et al) and chewing gum with probiotics, Scribante et al)
Materials and methods, the correct consort flow chart should be inserted, the one of 2025. How was the sample size calculated?
Results, very confusing, reformulate the tables highlighting more the significance dividing the secondary risk factors by categories
Discussion add in future objectives, the possibility of a multidisciplinary work between gnathologist orthodontist and dental hygienist to improve the quality of chewing and life of patients.
Conclusions, I would add the proactive action
Bibliography: add requested references
Author Response
I think the importance of the results obtained should be emphasized in the abstract because the conclusions are a bit weak.
Response: We thank the reviewer for your valuable advice. We emphasized “conclusion in Abstract” to clarify the significance of our study according to your suggestion (lines 28-30).
The keywords are few and not registered on MeSh, please update them.
Response: We thank the reviewer for your valuable advice. We have revised and added “Keywords” according to your suggestion (lines 31-32).
In the introduction I would do an overview of all the natural substances that can regulate the estrobolome even in the form of chewing gum (add reference of the estrobolome (Orrù et al) and chewing gum with probiotics, Scribante et al).
Response: We thank the reviewer for your valuable advice. Our study reports on the association between chewing status and SLD, not on the association between chewing gum and SLD. Therefore, it is difficult to comment in “Introduction” about chewing gum with probiotic because the point is blurred. On the other hand, since it was reported that estrogen was associated with SLD, we have added the sentences to “Discussion” according to your suggestion (lines 214-220, Reference 36-37).
Materials and methods, the correct consort flow chart should be inserted, the one of 2025. How was the sample size calculated?
Response: We thank the reviewer for your valuable advice. We have revised “Flowchart” according to your suggestion. In addition, our study is all-participant survey of all community residents who underwent medical checkups at Junpukai Health Maintenance Center in Okayama, Japan, and who were not diagnosed with SLD in the 2018 baseline survey. We have added the text to “Materials and methods” (lines 78-87, Figure 1).
Results, very confusing, reformulate the tables highlighting more the significance dividing the secondary risk factors by categories.
Response: We thank the reviewer for your valuable advice. As you indicated, we analyzed secondary risk factors categorically in logistic regression analysis. According to your suggestion, we have listed the only factors that were significantly different in multivariate logistic regression analyses in “table 4” because these are highlighted, and these factors are described in “Discussion” (Table 4).
Discussion add in future objectives, the possibility of a multidisciplinary work between gnathologist orthodontist and dental hygienist to improve the quality of chewing and life of patients.
Response: We thank the reviewer for your valuable advice. As you indicated, actively recommending dental visits to examinees with poor chewing status, treatment by orthodontists, and dental health guidance by dental hygienists may not only im-prove chewing status but also prevent the onset of SLD. We have added the text to “Discussion” and added “References” according to your suggestion (lines 228-232, Reference 40-41).
Conclusions, I would add the proactive action.
Response: We thank the reviewer for your valuable advice. As you indicated, aggressively recommending dental visits to participants with poor chewing status may not only improve their ability to chew well, but may also reduce the incidence of SLD. We have added the text to “Conclusions” according to your suggestion (lines 284-286).
Bibliography: add requested references.
Response: We thank the reviewer for your valuable advice. We have added “References” according to your suggestion (Reference 37).
Round 2
Reviewer 1 Report
Comments and Suggestions for Authors
Minor revision
Dyslipidemia is generally diagnosed by triglyceride level ≥ 150 mg/dL or HDL level ≤ 40 mg/dL. The authors should revise the description (line 107, 174, and Table 3) and re-analyze the data.
Author Response
Dyslipidemia is generally diagnosed by triglyceride level ≥ 150 mg/dL or HDL level ≤ 40 mg/dL. The authors should revise the description (line 107, 174, and Table 3) and re-analyze the data.
Response: Thank you for your sugesstion. As you pointed out, the values were misplaced. We have revised the text and “Table” according to your suggestion. (lines 105, 106, 111, 113, 177, 178, 187, 202, Table 1, 3).
Reviewer 3 Report
Comments and Suggestions for Authors
THE MANUSCRIPT HAS BEEN REVISED ALMOST IN EVERY PART, THE SECTION CONCERNING THE RECOMMENDED NATURAL SUBSTANCES IS MISSING (SCRIBANTE ET AL). THE REST IS FINE, CONGRATULATIONS TO THE AUTHOR
Author Response
THE MANUSCRIPT HAS BEEN REVISED ALMOST IN EVERY PART, THE SECTION CONCERNING THE RECOMMENDED NATURAL SUBSTANCES IS MISSING (SCRIBANTE ET AL).
Response: We thank the reviewer for your valuable advice. We have added the text and “References” according to your suggestion (lines 44-47, Reference 11, 12).